# The stability of covalent dative bond significantly increases with increasing solvent polarity

Rabindranath Lo [1,2], Debashree Manna[1,3], Maximilián Lamanec[1,4], Martin Dračínský [1], Petr Bouř[1], Tao Wu[1], Guillaume Bastien[1], Jiří Kaleta [1], Vijay Madhav Miriyala[1,2], Vladimír Špirko[1], Anna Mašínová[1], Dana Nachtigallová[1,5 ✉] & Pavel Hobza [1,5 ✉]

It is generally expected that a solvent has only marginal effect on the stability of a covalent bond. In this work, we present a combined computational and experimental study showing a surprising stabilization of the covalent/dative bond in $Me_3NBH_3$ complex with increasing solvent polarity. The results show that for a given complex, its stability correlates with the strength of the bond. Notably, the trends in calculated changes of binding (free) energies, observed with increasing solvent polarity, match the differences in the solvation energies ($\Delta E^{solv}$) of the complex and isolated fragments. Furthermore, the studies performed on the set of the dative complexes, with different atoms involved in the bond, show a linear correlation between the changes of binding free energies and $\Delta E^{solv}$. The observed data indicate that the ionic part of the combined ionic-covalent character of the bond is responsible for the stabilizing effects of solvents.

[1] Institute of Organic Chemistry and Biochemistry, Czech Academy of Sciences, Flemingovo náměstí 542/2, 16000 Prague, Czech Republic. [2] Regional Centre of Advanced Technologies and Materials, Czech Advanced Technology and Research Institute, Palacký University Olomouc, Křížkovského 511/8, 77900 Olomouc, Czech Republic. [3] Maulana Abul Kalam Azad University of Technology, West Bengal (formerly known as West Bengal University of Technology) Simhat, Haringhata, West Bengal 741249, India. [4] Department of Physical Chemistry, Palacký University Olomouc, 17. listopadu 12, 771 46 Olomouc, Czech Republic. [5] IT4Innovations, VŠB-Technical University of Ostrava, 17. listopadu 2172/15, 70800 Ostrava-Poruba, Czech Republic. ✉email: dana.nachtigallova@uochb.cas.cz; pavel.hobza@uochb.cas.cz

It is well known that the binding character between the two fragments **A** and **D** determines the stability of the (**A**–**D**) complex: the covalent bond (CB) complexes, realized by electron-sharing mechanisms, are generally more stable than the weaker non-covalent (NC) complexes. The dative bond (DB) complexes, also known as the donor–acceptor, coordinate-covalent, semi-polar, charge-transfer complexes, are bound by the specific type of a CB in which two electrons shared within the bond are provided by one fragment (donor, **D**) to the other (acceptor, **A**). A solvent further modifies the complex strength. While the stabilities of the former CB complex usually do not change significantly, the NC complexes are often considerably destabilized in the solvent.

The following equations can rationalize the effect of the solvent:

$$E_{stab} = E(A - D) - [E(A) + E(D)] \tag{1}$$

$$E_{stab}^{solv} = E(A - D)^{solv} - \left[E(A)^{solv} + E(D)^{solv}\right] \tag{2}$$

which describe the binding (stability) energies of the complex in the gas phase and solvent, respectively. The relative values of stabilization energies of isolated fragments **A** and **D** with respect to the complex (**A**–**D**) decide whether the solvent stabilizes/destabilizes the complex due to the different polarity. These values directly correlate with a particular complex's solvation energy ($E^{solv}$) in various solvents. For example, the stabilization of the ion-pair complexes gradually decreases with increasing solvent polarity[1], as the solvation of bare ions is larger than that of the neutral complex. Similar destabilization is observed in hydrogen-bonded and other NC complexes[2]. Intuitively, a larger solvation of the complex would require a significant charge redistribution in the complex as opposed to a smaller solvation of the isolated fragments; however, this was not observed in the majority of the NC complexes. A specific character of the DBs (see below) raises the question of the effect of the solvent on the stability of DB complexes.

The discussion on the DB goes back to Lewis's concept of base and acid[3] as species that share two electrons of the base to form a DB. This was followed by the work of Pauling[4], who described the DB as a "double bond" with one ionic and one covalent component. Further work by Haaland[5] proposed a distinction between the dative and CB complex with respect to its dissociation. In the case of the DB complex, the bond breaks heterolytically, yielding neutral, diamagnetic species, while the homolytic bond cleavage prevails in the case of the CB complex. Haaland discussed the nature of the DB for complexes which include main group metals. Further development, primarily done by Frenking, shows that the DB occurs much more frequently than initially thought in the main-group compounds, and it is a common feature of the s/p block atoms[6–14]. Relevant discussions on the nature and concept of the DB continue to appear in the literature[15,16].

The character of the DB, which combines the covalent and ionic character, is described by the wavefunction[17,18],

$$\Psi_{dative}\left(D^+ - A^-\right) = a\Psi_{covalent}(D - A) + b\Psi_{ionic}\left(D^+, A^-\right) \tag{3}$$

Since a charge transfer from the electron donor to the electron acceptor provides a measure of the ionic character, it will be used in further discussion. Electron transfer dynamics in the DB complexes have been observed experimentally employing spectroscopic studies with femtosecond resolution[18]. The character of DB complexes and their stabilities have been addressed in several computational studies[17,19–22].

The unusual behavior of DBs upon the surrounding changes of the environment is reflected by the sensibility of the bond distances when comparing the gas and solid states[23–25]. They sometimes differ by more than 1 Å, which contradicts the "normal" CB where the corresponding change is insignificant[19]. To our knowledge, the solvent effects on DB complex stability and the strength of the bond have not yet been discussed. However, their understanding, including the DB characterization in various solvents, is of great importance, and has led to growing interest in this type of bond. As discussed in a recent review (see ref. [6] and references therein), the concept of a DB has been addressed in connection to new classes of compounds.

The complex stabilization energy (correlated with the **A**–**D** bond dissociation energy ($D_e$)) and the strength of the **A**–**D** bond can characterize the stability of the complex. However, as discussed below, these properties do not always directly correlate[9,13,26].

The bond strength evaluation is not a trivial task, as it is not experimentally observable for polyatomic molecules. Thus, it can be obtained only via the evaluation of other quantities. It has been shown in several cases[9,27] that neither the bond dissociation energy ($D_e$), the vibrationally corrected zero-point energy ($D_0$), nor the activation energy of the bond-breaking process to fragments **A** and **D** ($E^{act}$) reflect their bond strength/energy. This is due to possible modifications of the electronic states of the fragments in the (**A**–**D**) and separated **A** and **D** states. The widely accepted assumption that stronger bonds are shorter than weaker ones is not generally correct.

Moreover, due to the delocalization of the normal coordinates, traditionally used vibrational analyses do not provide direct information on the stretching force constants and fundamental frequency of a particular bond, and consequently, its strength[9,27]. To overcome this problem, Konkoli, Larsson and Cremer[28–31] transformed the delocalized normal modes into new vibrational modes ("adiabatic internal modes"). Alternatively, one can rely on the so-called HBJ approach[32], which is based on introducing a non-rigid reference configuration of the molecular atoms which essentially follows the respected vibrational motion. This approach accounts exactly for anharmonicity of the motion and also allows to account easily for the most important kinematic interaction terms.

The combined ionic-covalent character of the DB indicates that the solvent might significantly affect the strength and stability of the DB complexes. Recently, we investigated the DB complexes between electron acceptors ($C_{60}$, $C_{20}$, $C_{70}$, cyclo[n]carbons, graphene, and single-wall nanotubes) and electron donors (piperidine, piperidine dimer, phosphines) using the tools of computational chemistry and several experimental techniques to characterize the DBs in terms of the charge transfer and stabilization energy[33–38]. The studies are also performed in a solvent, providing surprising findings on the solvent effects that motivated a thorough survey of this phenomenon.

The current paper reports on a combined vibrational Raman and NMR spectroscopic and computational investigation of the solvent effect on the complex stability and bond strength in the $Me_3NBH_3$ complex, a modification of a prototype covalent dative $NH_3$–$BH_3$ complex. The vibrational spectra provide information on the character of DBs in terms of vibration frequencies related to DB force constants. In the NMR experiment, the indirect spin–spin coupling (*J*-coupling) is mediated by involved electrons. It is thus related to chemical bonding (in contrast to direct coupling, which is a through-space interaction)[39] and is used to discuss the changes in B–N distances. However, the magnitude of the *J*-coupling cannot be directly correlated to the bond strength. The solvent's effect is further investigated through computational studies for complexes between larger electron acceptors ($C_{18}$, $C_{60}$ and its derivatives, and $C_{70}$) and electron donors piperidine (pip) and tris(1-pyrrolidinyl)phosphine (P(pyrr)$_3$).

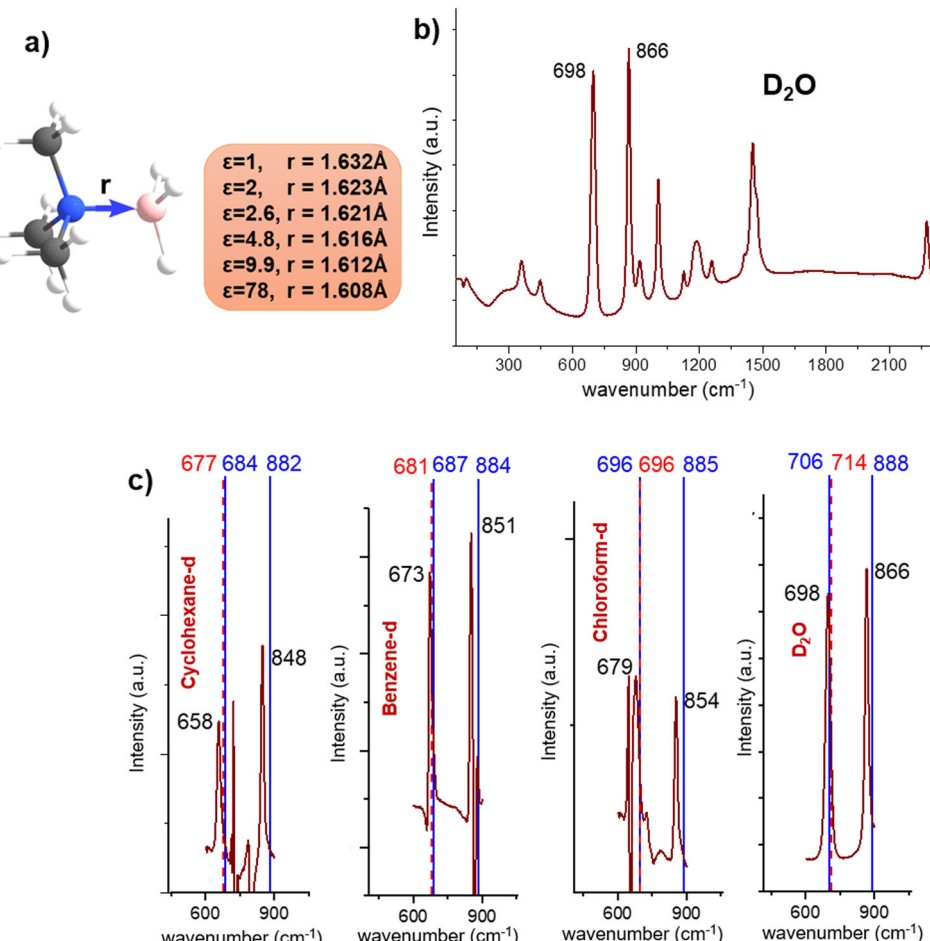

**Fig. 1 The calculated and observed structural and spectroscopic data of Me₃NBH₃. a** The optimized geometry of Me₃NBH₃. The B–N distances (*r*) are given in the gas phase (ε = 1) and various solvents, cyclohexane (ε = 2.0), carbon disulfide (ε = 2.6), chloroform (ε = 4.8), *o*-dichlorobenzene (ε = 9.9), and water (ε = 78.0); **b** the Raman spectra of Me₃NBH₃ measured in D₂O; **c** the relevant part of the Raman spectra ranging 450–1050 cm⁻¹ in various solvents. The PBE0-D3/def2-QZVP calculated harmonic (in blue color) and anharmonic (in red color) frequencies are shown (C: gray, N: blue, H: white, B: pink).

## Results and discussion

**Me₃NBH₃.** Figure 1a displays the structure of Me₃NBH₃ with the optimized B–N distance obtained in the gas phase (ε = 1.0) and in various solvents, including cyclohexane (ε = 2.0), carbon disulfide (ε = 2.6), chloroform (ε = 4.8), *o*-dichlorobenzene (ε = 9.9), and water (ε = 78.0). Similar to findings discussed by Bühl et al.[40] and Jonas et al.[19], the distance gradually shortens upon increasing solvent polarity. Notice that MD simulations in explicit solvents CHCl₃ and CS₂ (see the SI) fully confirmed this finding.

As stated in ref. [40], the changes in the geometry in the solvent influence the chemical shift observed in the NMR spectra. However, it is difficult to distinguish these geometry-related chemical-shift changes from those caused by the magnetic shielding of the solvent molecules. Therefore, we rely on indirect NMR coupling rather than chemical shift. Supplementary Figure 1 displays the NMR spectrum of Me₃NBH₃ obtained in CD₃CN; the values of indirect one-bond B–N and B–H couplings in various solvents with dielectric constants ranging from 2 to 78 are presented in Supplementary Table 1. The information obtained from the B–H couplings is only indirectly connected to the observed DB. However, the almost linear correlation between the measured *J*(N–B) and *J*(B–H) values (Supplementary Fig. 1) justifies using both data for further discussion. Table 1 displays the values of experimental and calculated spin–spin couplings

related to the B–N bond in selected solvents. The calculations underestimate the observed values, with the error in the range of 0.6–0.9 Hz, with the exception of a more significant error of 2.4 Hz in water. Despite these differences, both sets of results show increased spin–spin coupling with increasing solvent polarity. Further calculations show that the Fermi contact couplings are the main contributors to the total spin–spin couplings, with the same dependence on the solvent polarity (Table 1). Both the total spin–spin couplings and Fermi contact parts linearly correlate with the calculated bond distances obtained in the various solvents (Supplementary Fig. 2).

Figure 1b shows the representative Raman spectrum of Me₃NBH₃ measured in D₂O with the two most intensive peaks at 698 and 866 cm⁻¹, mainly due to B–N and C–N stretching vibrations. The relevant part of the Raman spectra in the range 450–1050 cm⁻¹ of Me₃NBH₃ in cyclohexane, benzene, chloroform, and water and their assignments are given in Fig. 1c, Table 2 and Supplementary Table 2. The potential energy distribution calculations assign the lower frequency peaks mainly to the N–B stretching (65–69%), while the higher frequency peaks have small contributions (12–18%) and are primarily due to the C–N stretching vibrations (67–75%). The calculations of vibrational spectra within the harmonic approximation overestimate the frequencies more significantly for high-frequency peaks (by 22–34 cm⁻¹); however, they are more consistent than

**Table 1 The experimental B-N indirect coupling ($^1J$(B-N)$_{exp}$), calculated B-N distance ($R_{B-N}$), B-N indirect coupling ($^1J$(B-N)$_{calc}$, deviation from the experiment in parenthesis) and Fermi contact coupling ($^1J_{FC}$(B-N)$_{calc}$) of Me$_3$NBH$_3$.**

| Solvent | $\varepsilon$ | $^1J$(B-N)$_{exp}$ Hz | $^1J$(B-N)$_{calc}$ Hz | $^1J_{FC}$(B-N)$_{calc}$ Hz | $R_{B-N}$ Å |
|---|---|---|---|---|---|
| Cyclohexane | 2.0 | 5.3 | 4.4 (0.9) | 4.13 | 1.623 |
| Benzene | 2.3 | 6.2 | 4.6 (0.6) | 4.31 | 1.622 |
| Acetone | 20.5 | 6.8 | 5.9 (0.9) | 5.69 | 1.610 |
| Acetonitrile | 35.7 | 7.1 | 6.0 (1.1) | 5.77 | 1.609 |
| Water | 78.0 | 8.5 | 6.1 (2.4) | 5.85 | 1.608 |

The values are calculated at the PBE0/def2-QZVP level.

**Table 2 Observed fundamental frequencies ($\nu_{obs}$), harmonic frequencies ($\nu_{harm}$), anharmonic ($\nu_{anharm}$), and scaled anharmonic ($\nu_{anharm}^{scal}$) vibrational frequencies calculated using the HBJ approach with rigid-bender reduced masses ($\mu_{ss}$) and scaled masses ($\mu_{ss}/1.0465$), respectively (frequencies given in cm$^{-1}$, the scaling factor 1.0465 chosen to reproduce the experimental B-N fundamental frequency pertaining to water).**

| Solvent | $\varepsilon$ | $\nu_{obs}$ | PED | $\nu_{harm}$ ($\Delta\nu$) | $\nu_{anharm}$ ($\Delta\nu$) | $\nu_{anharm}^{scal}$ ($\Delta\nu$) | $\mu_{ss}$ |
|---|---|---|---|---|---|---|---|
| Gas phase | 1.0 | | | 670 | 656 | 642 | 0.12854 |
| Cyclohexane | 2.0 | 658 | 69 (18) | 684 (26) | 677 (19) | 663 (5) | 0.12843 |
| Benzene | 2.3 | 673 | 69 (19) | 687 (14) | 681 (8) | 666 (-7) | 0.12840 |
| Chloroform | 4.8 | 679 | 67 (22) | 696 (17) | 696 (17) | 681 (2) | 0.12827 |
| Water | 78.0 | 698 | 65 (26) | 706 (8) | 714 (16) | 698 | 0.12817 |

The deviation from the experiment ($\Delta\nu$) and assignment of B-N stretching with the potential energy distributions, PED% (C-N stretching in parenthesis) of the normal vibrational modes of Me$_3$NBH$_3$ in various solvents. The values are calculated at the PBE0-D3/def2-QZVP level.

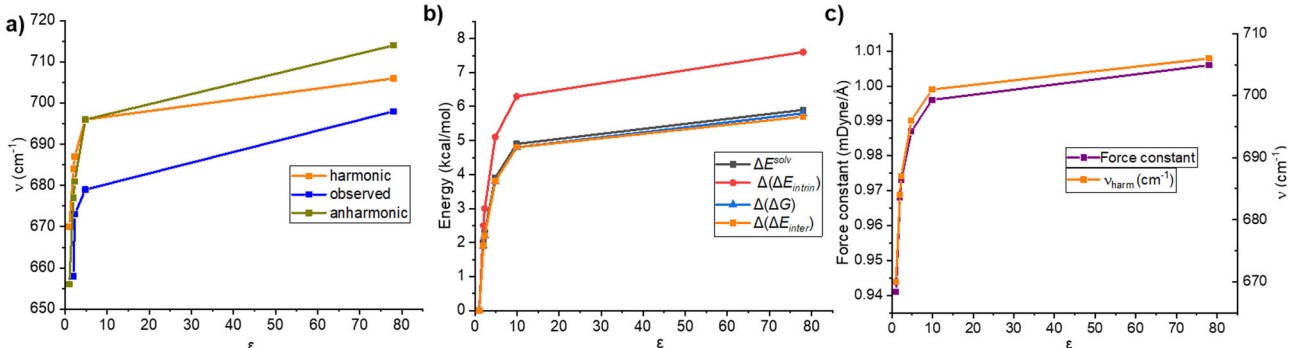

**Fig. 2 The correlation between the observed and calculated properties of Me$_3$NBH$_3$ and the solvent polarity expressed by the dielectric constant $\varepsilon$.** **a** The observed and calculated an/harmonic frequencies versus $\varepsilon$; **b** the calculated $\Delta(\Delta G)$, $\Delta(\Delta E_{inter})$, $\Delta(\Delta E_{intrin})$, $\Delta E^{solv}$ versus $\varepsilon$; and **c** the force constant and vibration frequency of B-N stretching mode versus $\varepsilon$. The values are calculated at the PBE0-D3/def2-QZVP level.

low-frequency peaks, with errors in the range of 8–26 cm$^{-1}$. Note that the effect of anharmonicity is the largest for H$_2$O. In addition, we observed a similar decline of calculated harmonic frequencies from the experiment, as found in the case of $J$(B–N) values. The dependence of the observed and the calculated harmonic and anharmonic frequencies on the solvent polarity is illustrated in Fig. 2, showing a gradual blue-shift with increasing solvent polarity with the largest changes in the region up to $\varepsilon = 4.8$. Both the anharmonic and harmonic calculated frequencies thus provide reliable values in terms of the solvation trends and can be used for further discussion.

Table 3 summarizes the results of the calculated complex stabilities of Me$_3$NBH$_3$ in various solvents. Following the discussion on the characterization of the bond strength (see "Introduction"), we characterized the complex in terms of the interaction energy ($\Delta E_{inter}$), Gibbs free energy ($\Delta G$) calculated at $T = 298$ K, and intrinsic energy ($\Delta E_{intrin}$), which neglects deformation energies of the isolated monomers. The results show considerable stability of the complex even in the gas phase ($\Delta E_{inter}$ and $\Delta G$ of $-41.8$ and $-24.9$ kcal/mol, respectively). The

solvent further stabilizes the complex with respect to its polarity as presented for observed and calculated vibrational frequencies. The differences are most significant for the lower polarity solvents and become smaller once the solvent polarity reaches $\varepsilon \sim 10$. Table 3 also shows the values of $\Delta E^{solv}$, i.e., the differences between the solvation energies of complexes and isolated monomers. Following the discussion in the Introduction, this term is expected to provide specific information on the solvent polarity effect on the complex stability. Figure 2 illustrates that an exact match exists between the changes in the binding free energies ($\Delta(\Delta G)$), intrinsic interaction energies ($\Delta(\Delta E_{intrin})$) and changes in solvation energies, ($\Delta E^{solv}$). Thus, the latter can be used as a reliable indicator of the solvent effects on the dative-bond complex stability. Increasing $\Delta E^{solv}$ values can be rationalized by the inspection of the $\Psi_{dative}$ wavefunction, which is a linear combination of $\Psi_{covalent}$ and $\Psi_{ionic}$ (Eq. 3). With the increasing solvent polarity, the corresponding energy term of the covalent contribution changes only negligibly, while that of the ionic contributions significantly stabilizes. The values of the relative charge transfer ($Q_{rel}$, Table 3) and their almost linear

**Table 3 The interaction energies ($\Delta E_{inter}$, kcal/mol), thermodynamics characteristics ($\Delta G$, kcal/mol), differences in the solvation energies ($\Delta E^{solv}$, kcal/mol), intrinsic energies ($\Delta E_{intrin}$, kcal/mol), force constants ($k^c$, mDyne/Å), and the relative charge transfer ($Q_{rel}$) of $Me_3NBH_3$ in various solvents.**

| Solvent | $\varepsilon$ | $\Delta E_{inter}$ | $\Delta E_{intrin}$ | $\Delta G$ | $\Delta E^{solv}$ | $k^c$ | $Q_{rel}$[a] |
|---|---|---|---|---|---|---|---|
| Gas phase | 1.0 | −41.8 | −57.6 | −24.9 | 0.0 | 0.941 | 0.0 |
| Cyclohexane | 2.0 | −43.7 | −60.1 | −26.8 | 2.0 | 0.968 | 0.019 |
| Chloroform | 4.8 | −45.6 | −62.7 | −28.7 | 3.9 | 0.987 | 0.032 |
| O-dichlorobenzene | 9.9 | −46.6 | −63.9 | −29.7 | 4.9 | 0.996 | 0.038 |
| Water | 78.0 | −47.5 | −65.2 | −30.7 | 5.9 | 1.006 | 0.043 |

The values are calculated at the PBE0-D3/def2-QZVP level.
[a]The values relate to $Q$ in the gas phase ($Q = 0.347$).

correlation with $\Delta E^{solv}$ (Supplementary Fig. 3) nicely illustrate the solvent's effect on the ionic part of the bond.

According to the previous discussion, the stability of the complex in general does not reflect the strength of the bond connecting two fragments. Therefore, it is interesting to investigate whether complex stability and strength of the connecting bond correlate for this specific case. Importantly for this discussion, small changes in the Wiberg bond indexes within the whole range of $\varepsilon$, with the values of 0.58–0.62, indicate that the bond character does not change in investigated solvents. This provides arguments to discuss the correlation of the stability of DB complexes and the relevant bond strengths in terms ($\Delta(\Delta G)$, $\Delta(\Delta E_{inter})$) or $\Delta E^{solv}$ with changes in the intrinsic energies ($\Delta(\Delta E_{intrin})$) and force constant of the low-frequency vibrations, also listed in Table 3. Arguments for the validity of the use of ($\Delta(\Delta E_{intrin})$) values are based on the assumption that the low-frequency vibrations are mainly due to the B–N stretching with an almost constant contribution (65–67%, Table 2) to the normal mode motion. Figure 2 illustrates the dependences of all these characteristics and vibration frequency on the solvent polarity, showing almost exact matches in their trends. This observation allows us to conclude that there is a direct correlation between changes in the strength of the DB and the complex stability due to the solvation for a given DB complex. Notably, both values correlate with the $\Delta E^{solv}$ value, making it a reliable indicator of changes in complex stabilities with changes in solvent polarities.

**Other DB complexes**. Table 4 displays the values of $\Delta G$ calculated in the gas phase and o-dichlorobenzene ($\varepsilon = 9.9$), $\Delta(\Delta G)$ and $\Delta E^{solv}$ values for various DB complexes (see Supplementary Fig. 4). They were selected to include different types of the DB bond, i.e., $N \rightarrow B$, $N \rightarrow C$, $P \rightarrow B$, and $P \rightarrow C$ bonds. The complexes represent a relatively large variation of their structures, including smaller systems, such as the prototype DB complex $H_3B$-$NH_3$, and large systems, e.g., $C_{60}(CN)_{18}\ldots P(pyrr)_3$. The investigated complexes represent a large variety in terms of the polarity of their fragments. We thus considered non-polar electron donors $C_{60}$ as well as their highly polar modifications $C_{60}(CN)_4$, $C_{60}(CN)_{18}$, and $C_{60}F_{18}$. Regarding their stabilities based on Gibbs's free energies, the selection covered unstable complexes, e.g., $C_{60}\ldots pip_2$, with $\Delta G_{gas} = 4.3$ kcal/mol up to complexes with higher stabilities, e.g., $H_3B$-$PMe_3$ with $\Delta G_{gas} = -31.8$ kcal/mol. In all cases, the solvent stabilized the complex. The values of $\Delta E^{solv}$ provided a very similar picture. The functionalization of the electron donor ($NH_3$) with the electron-donating $CH_3$ group leads to a larger complex stabilization in both gas phase and o-DCB. This effect is more pronounced in the gas phase. The functionalization of the electron acceptor ($BH_3$) with electron-withdrawing F atoms destabilizes the complex in both environments. The observed linear relation between $\Delta(\Delta G)$ and $\Delta E^{solv}$ of a particular bond type ($N \rightarrow B$, $N \rightarrow C$, $P \rightarrow C$, see

**Table 4 The thermodynamic characteristics calculated in the gas phase ($\Delta G_{gas}$, kcal/mol), o-dichlorobenzene ($\Delta G_{o-DCB}$, kcal/mol), their differences $\Delta(\Delta G)$, kcal/mol), and differences in the solvation energies ($\Delta E^{solv}$, kcal/mol) of selected dative bond complexes.**

| | $\Delta G_{gas}$ | $\Delta G_{o-DCB}$ | $\Delta(\Delta G)$ | $\Delta E^{solv}$ |
|---|---|---|---|---|
| $N \rightarrow B$ | | | | |
| $H_3B$-$NH_3$ | −19.5 | −27.5 | 8.0 | 8.4 |
| $H_3B$-$NMe_3$ | −24.9 | −29.7 | 4.8 | 4.9 |
| $F_3B$-$NH_3$ | −9.0 | −19.1 | 10.1 | 11.3 |
| $F_3B$-$NMe_3$ | −13.6 | −19.0 | 5.4 | 6.0 |
| $N \rightarrow C$ | | | | |
| $C_{60}\ldots pip_2$ | 4.3 | 0.9 | 3.4 | 4.2 |
| $C_{70}\ldots pip_2$ | 1.8 | −2.2 | 4.0 | 4.4 |
| $C_{18}\ldots pip$ | −1.6 | −15.3 | 13.7 | 15.5 |
| $P \rightarrow B$ | | | | |
| $H_3B$-$PMe_3$ | −31.8 | −36.1 | 4.3 | 4.5 |
| $P \rightarrow C$ | | | | |
| $C_{60}\ldots P(pyrr)_3$ | −0.7 | −5.2 | 4.5 | 6.1 |
| $C_{60}F_{18}\ldots P(pyrr)_3$ | −6.1 | −10.1 | 4.0 | 5.5 |
| $C_{60}(CN)_4\ldots P(pyrr)_3$ | −6.0 | −9.3 | 3.3 | 4.7 |
| $C_{60}(CN)_{18}\ldots P(pyrr)_3$ | −16.3 | −16.5 | 0.2 | 0.7 |

The $N \rightarrow B$ and $P \rightarrow B$ complexes are calculated at the PBE0-D3/def2-QZVP, whereas C18 complex is computed at the ωB97XD/def2-TZVPP level. The other complexes are calculated at the PBE0-D3BJ/def2-TZVPP level.

Supplementary Fig. 5) indicates that for a given DB, the values of $\Delta E^{solv}$ follow almost the same trends as $\Delta(\Delta G)$, and thus can serve as a direct indicator of the solvent effect on the DB complexes.

In summary, it is generally assumed that the complexes decrease their stability in the solvent compared to the gas phase. These changes are small in the complexes bound via "normal" CBs, while that can be significant in NC complexes. The specificity of the DB, which combines covalent and ionic characters, connected with a substantial charge transfer character, opens the question of the solvent polarity of DB complexes. Our calculations in different solvent media show their surprising effects on the DB character:

- All complexes increased their stability in solvent compared to the gas phase. Moreover, for the same bond types ($N \rightarrow C$, $N \rightarrow B$, $P \rightarrow B$, and $P \rightarrow C$), increasing stabilities correlate linearly with the differences in solvation energies of the complexes and isolated fragments.
- The combined experimental and computational studies performed for $Me_3NBH_3$ show that increasing polarity of the solvent correlates not only with the larger complex stability, evaluated using interaction energies and Gibb's free energies, but also with a stronger DB, evaluated through intrinsic energies, vibrational frequencies, and

force constant. Furthermore, the relative values of the charge transfer correlate with an increasing ionic character of the complex in polar solvents connected to a smaller N–B separation, which stabilizes the energy corresponding to the ionic part of the bond.

## Methods

**Calculations**. The structures of the DB complexes were optimized at the DFT-D level, employing the PBE0-D3 functional[41,42], using the def2-QZVP basis set[43] for $H_3NBH_3$, $Me_3NBH_3$, $H_3NBF_3$, $Me_3NBF_3$ and $Me_3PBH_3$ and def2-TZVPP basis set for other systems, respectively. The reliability of the DFT-D approach was justified by a comparison of binding energies calculated at the CCSD(T)/cc-pVTZ level for $H_3NBH_3$ and $Me_3NBH_3$. All CCSD(T) results of interaction energies were corrected for the basis set superposition error. Gibbs's free energies were calculated at the temperature $T = 298$ K.

The CCSD(T)/cc-pVTZ and DFT-D binding energies of $NH_3BH_3$ equal 33.0 and 35.3 kcal/mol, respectively, both in good agreement with the previously reported benchmark value of 31.6 kcal/mol calculated with the CCSD(F12)(T)/CBS approach[16]. The T1 diagnostic value of 0.008 indicates that the complex is predominantly single reference in nature.

The methyl substitution of nitrogen substantially increases its donor ability, increasing binding energy $Me_3NBH_3$. The binding energy is 41.9 kcal/mol and 41.8 kcal/mol at the CCSD(T)/cc-pVTZ and DFT-D/QZVP, respectively, justifying the use of the latter approach. Note also that the present DFT-D approximation was shown to provide reasonably accurate binding energies and geometries for DAT20 dataset containing 20 DB complexes[16].

The total interaction energies are calculated from the fully optimized structures of complex and fully optimized isolated monomers using the following equation:

$$\Delta E_{\mathrm{inter}} = E_{\mathrm{complex}} - E^i_{\mathrm{monomer1}} - E^i_{\mathrm{monomer2}} \qquad (4)$$

The intrinsic interaction energy ($\Delta E_{\mathrm{intrin}}$), calculated as a difference between energy of the optimized complex and the sum of the energies of subsystems with geometries taken from the optimized complex geometry.

$$\Delta E_{\mathrm{intrin}} = E_{\mathrm{complex}} - E_{\mathrm{monomer1}} - E_{\mathrm{monomer2}} \qquad (5)$$

The solvent effects were included using the continuous model[44], in which the specific solvent is characterized by its dielectric constant $\varepsilon$. The calculations were performed in the gas phase ($\varepsilon = 1.0$) and various solvents, cyclohexane ($\varepsilon = 2.0$), benzene ($\varepsilon = 2.3$), carbon disulfide ($\varepsilon = 2.6$), chloroform ($\varepsilon = 4.8$), 1,2-dichlorobenzene ($\varepsilon = 9.9$), and water ($\varepsilon = 78.0$). The solvation energy ($E^{\mathrm{solv}}$) of the system in a particular solvent is calculated as a difference between the energy of the optimized system in that solvent and the energy of the same geometry in the gas phase. Molecular dynamics studies with the discrete solvent model were performed for selected systems to check for the reliability of the continuous solvent model. In particular, the MD simulations were performed for $Me_3NBH_3$ in vacuo and embedded in a cluster of 49 molecules of chloroform and 49 molecules of carbon disulfide.

The bond properties (Wiberg bond indices and charge transfer) were evaluated using the NBO analyses[45]. Although the partial charges used for the charge transfer description are not well defined, and their values depend on the method, they can be used to obtain relative values in different solvents.

**Vibrational frequencies**. The sought vibrational energies associated with the B–N stretching motion are obtained by solving the Schrödinger equation for the following vibrational HBJ Hamiltonian[32],

$$H_{\mathbf{con}} = -\frac{1}{2}\mu_{ss}J_s^2 + \frac{1}{2}\left(J_s\mu_{ss}\right)J_s + \frac{1}{2}\mu^{\frac{1}{4}}\left\{J_s\mu_{ss}\mu^{-\frac{1}{2}}\left[J_s\mu^{\frac{1}{4}}\right]\right\} + V(s), \qquad (6)$$

where $J_s = -i\hbar\left(\frac{d}{ds}\right)$, $\mu_{ss}(s)$ is the B–N stretching component of the tensor that is the inverse of the $4 \times 4$ generalized molecular inertia tensor, $\mu$ is the determinant of the matrix $[\mu_{\alpha,\beta}]$ ($\alpha, \beta = x, y, z, s; x, y, z$ being the Cartesian atomic coordinates in the molecular-fixed-axis system), and $V(s)$ is the B–N energy minimum path potential. As seen in Supplementary Fig. 6, all the atomic coordinates of $Me_3NBH_3$ used in this study as a model system exhibit fairly linear dependence on the stretching distortion; for further discussion see SI material.

**Nuclear magnetic resonance (NMR) spectroscopy**. NMR is a universal analytical method that provides atomic-level insights into molecular structure. Indirect spin–spin coupling (J-coupling) is one of the most important NMR parameters. It is tightly related to chemical bonding because it is mediated by electrons (in contrast to direct coupling due to a through-space interaction)[39]. We prepared a $^{15}N$ labeled trimethylamine–borane complex and measured indirect one-bond B–N and B–H couplings in various solvents with dielectric constants ranging from 2 to 78 (Supplementary Table 1). Supplementary Figure 1 displays an example spectrum measured in acetonitrile. The B–N coupling values clearly increase with the increasing polarity of the solvent, while the B–H couplings have the opposite trend.

**Raman spectroscopy**. We used Raman spectra to monitor the DB force constants of the model compound, as reflected in the vibrational frequencies. The same solutions of the trimethylamine–borane complex as for NMR were measured on a spectrometer or a microscope, using the 532 nm laser excitation. Some signals were too weak or masked by fluorescence. However, in most cases, vibrational bands originated to a great extent in the B–N stretching (within ~700–1500 $cm^{-1}$) could be extracted; the spectra are exemplified in Supplementary Figs. 7–11. The B–N stretching band frequencies exhibited significant solvent dependence; for example, the strongest 866 $cm^{-1}$ vibration in $D_2O$ moved down to 848 $cm^{-1}$ in cyclohexane.

## Data availability

All data resulting from the experimental and computational studies of this work are included within this Article and the Supplementary Information.

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

## Acknowledgements

This work was supported by the Czech Science Foundation, projects 19-27454X (to P.H. and D.N.), 20-10144S (to P.B.), 22-15374S (to M.D.), and 20-13745S (to J.K.); by Palacký University, the Internal grant association, the project IGA_PrF_2022_019 (to M.L.). D.M. would like to acknowledge the Department of Science and Technology, India for funding (DST/INSPIRE/04/2019/000065).

## Author contributions

P.H. and D.N. supervised the project. R.L., D.M., M.L., A.M., and V.M.M. carried out the quantum chemical calculations. V.M.M. carried out molecular dynamics simulations. V.S. performed the anharmonic frequency calculations. M.D., P.B., T.W., G.B., and J.K. performed the experiments. M.D. and P.B. analyzed the experimental data. P.H. and D.N. jointly interpreted all data and wrote the manuscript. All authors discussed the results and commented on the manuscript.

## Competing interests

The authors declare no competing interests.
