## [Peer Review File · Nature Communications]

REVIEWERS' COMMENTS

Reviewer #1 (Remarks to the Author):

This work reports results of fundamental research on the influence of solvents on the stabilities of compounds featuring dative bonds. High level theoretical calculations in combination with Raman and NMR spectroscopy experiments have been used to study the Me₃NBH₃ complex, which shows a surprising increase in stability as solvent polarity is raised. The authors argue, soundly, that the wavefunction for the covalent component of the dative bond does not change appreciably with increasing solvent polarity but that of the ionic portion does. Several other examples featuring different types of dative bonds are also reported in the study. This dependence is unusual and, given the widespread occurrence of dative bonds, can be consequential in determining functions of such bonds in solvent media. Publication is recommended after minor revisions noted below.

- (a) Provide T1 diagnostic values for CCSD(T) calculations to check for multireference character in the wavefunction.
- (b) The isotopically labeled ¹⁵N sample contains CD₃ groups instead of CH₃ in Scheme S1. But the caption in Figure S1 says ¹⁵N-labeled Me₃NBH₃. Please clarify the use of deuterium labeling to avoid confusion.
- (c) Include methods and basis sets in Table/Figure captions.
- (d) In several places the language was a bit hard to follow so please reword for clarity where necessary.
- (e) Minor point: The top panel in Figure S7 needs to have the formula format for Me₃NBH₃.

Reviewer #2 (Remarks to the Author):

The manuscript of Lo et al. discusses the stability of dative bonds of molecules in solution with respect to the polarity of the solvent. The project nicely combines experiment with theory in order to prove the initial hypothesis of the authors, i.e. a more polar solvent results to a stronger dative bond. However, I am not so sure if this discovery justifies the publication of this work as a "communication". The authors discuss extensively a simple molecular system (Me₃NBH₃) at both experimental and computational level, and they extrapolate their finding with additional

computations on other molecular species with variable complexity. Here are my comments that the authors should address before this article is considered for publication:

- Since the authors have considered both ammonia borane and Me_3NBH_3 in this study, it would be interesting to examine what is the effect of the substituted species in the bond strength under different solvents. The discussion can be extended to the BH_3 and BF_3 species.
- In Table 4, is there any correlation between ΔG of solvation and the $\Delta\Delta G$ (difference between gas phase and in solution)?
- Have the authors used the Becke-Johnson damping to the D3 dispersion correction?
- What is the reason of such a large deviation between the CCSD(T)/cc-pVTZ and the CCSD(T)/CBS that is mentioned in the Methods section?
- Please fix typo CCSDT(T) on page 13

Reviewer #3 (Remarks to the Author):

The manuscript presents experimental and computational data elucidating the effect of solvation on the stability and the bond strength of dative bonds. The authors find that both stability and bond strength for dative bond increase with the dielectric constant of the solvent. This is an important finding that does not seem to be known. It differs strongly from the behavior of normal covalent bonds. It is laudable that the authors make a clear distinction between the bond energy or stability and the bond strength which is independent of the fragments.

The paper should be published after a revision addressing these issues:

(1) It is rightfully pointed out that "bond strength" cannot be determined directly, and is also not very well defined. They also point out that stability and bond strength do not necessarily correlate. Unfortunately, the manuscript is not really clear on which measures are used to assess bond strength. One good measure is the force constant of the bond, or the frequency, which is found in Table 2. The NMR J-coupling is the first measure given in Tab. 1. The authors write (p.4) "It thus reflects chemical bonding". Do they mean: "reflects bond strength"? Is there further evidence for this relation? If not what conclusion can be drawn from the NMR data?

In the conclusion, the "intrinsic energy" is stated as a measure for "bond strength". This is not elaborated on in the Introduction or Results section.

(2) The calculated data from Table 3 are very important for this work. Unfortunately, one can only guess from the Methods section what calculations are behind Table 3.

(3) The English could be improved

Authors' replies to the Reviewers' remarks and questions

Referee: 1

This work reports results of fundamental research on the influence of solvents on the stabilities of compounds featuring dative bonds. High level theoretical calculations in combination with Raman and NMR spectroscopy experiments have been used to study the Me₃NBH₃ complex, which shows a surprising increase in stability as solvent polarity is raised. The authors argue, soundly, that the wavefunction for the covalent component of the dative bond does not change appreciably with increasing solvent polarity but that of the ionic portion does. Several other examples featuring different types of dative bonds are also reported in the study. This dependence is unusual and, given the widespread occurrence of dative bonds, can be consequential in determining functions of such bonds in solvent media. Publication is recommended after minor revisions noted below.

Reviewer's Point #1:

Provide T1 diagnostic values for CCSD(T) calculations to check for multireference character in the wavefunction.

Response:

T1 diagnostic values for CCSD(T) calculation are now reported in the revised manuscript. The T1 diagnostic values suggest the systems are predominantly single reference in nature. We have included the following sentence in the revised manuscript (section Methods, pg 7) *"The T1 diagnostic value of 0.008, indicates that the complex is predominantly single reference in nature."*

Reviewer's Point #2:

The isotopically labelled ¹⁵N sample contains CD₃ groups instead of CH₃ in Scheme S1. But the caption in Figure S1 says ¹⁵N-labeled Me₃NBH₃. Please clarify the use of deuterium labelling to avoid confusion.

Response: We have corrected the caption in Figure S1.

Reviewer's Point #3:

Include methods and basis sets in Table/Figure captions.

Response:

In the revised version we have incorporated methods and basis sets in Table/Figure captions in the main text and SI.

Reviewer's Point #4:

In several places the language was a bit hard to follow so please reword for clarity where necessary.

Response:

The language was corrected by the native English-speaking chemist.

Reviewer's Point #5:

Minor point: The top panel in Figure S7 needs to have the formula format for Me₃NBH₃.

Response:

The formula format for Me₃NBH₃ is now incorporated in the top panel in Figure S7.

Referee: 2

The manuscript of Lo et al. discusses the stability of dative bonds of molecules in solution with respect to the polarity of the solvent. The project nicely combines experiment with theory in order to prove the initial hypothesis of the authors, i.e. a more polar solvent results to a stronger dative bond. However, I am not so sure if this discovery justifies the publication of this work as a "communication". The authors discuss extensively a simple molecular system (Me₃NBH₃) at both experimental and computational level, and they extrapolate their finding with additional computations on other molecular species with variable complexity. Here are my comments that the authors should address before this article is considered for publication.

Reviewer's Point #1:

Since the authors have considered both ammonia borane and Me₃NBH₃ in this study, it would be interesting to examine what is the effect of the substituted species in the bond strength under different solvents. The discussion can be extended to the BH₃ and BF₃ species.

Response:

We have now included the discussion regarding the effect of the substituted species in the bond strength under different solvents in the current version of the manuscript. We have included the following sentence in the revised manuscript, pg 6.

"The functionalization of the electron donor (NH₃) with electron-donating CH₃ group leads to a larger complex stabilization in both gas phase and o-DCB. This effect is more pronounced in the gas phase. The functionalization of the electron acceptor (BH₃) with electron-withdrawing F atoms destabilizes the complex in both environments."

Reviewer's Point #2:

In Table 4, is there any correlation between delta G of solvation and the delta delta G (difference between gas phase and in solution)?

Response:

Positive values of ΔG^{solv} (ΔE^{solv} in Table 4) indicate increasing complex stabilities in the polar solvents as predicted by $\Delta(\Delta G)$. This observation is documented in Fig S5 and discussed in Results and Discussion (pg 7)

“The observed linear relation between $\Delta(\Delta G)$ and ΔE^{solv} of a particular bond type ($N \rightarrow B$, $N \rightarrow C$, $P \rightarrow C$, see Supplementary Fig. 5) indicates that for a given dative bond, the values of ΔE^{solv} follow almost the same trends as $\Delta(\Delta G)$, and thus can serve as a direct indicator of the solvent effect on the dative bond complexes.”

Reviewer’s Point #3:

Have the authors used the Becke-Johnson dumping to the D3 dispersion correction?

Response: We have used Becke-Johnson dumping to the D3 dispersion correction for few complexes in Table 4 as indicated in the caption.

Reviewer’s Point #4:

What is the reason of such a large deviation between the CCSD(T)/cc-pVTZ and the CCSD(T)/CBS that is mentioned in the Methods section?

Response:

The difference of 1.4 kcal/mol (~ 5 %) is quite usual due to a relatively small cc-pVTZ basis set.

Reviewer’s Point #5:

Please fix typo CCSDT(T) on page 13

Response:

Typo has been corrected in the revised version.

Referee: 3

The manuscript presents experimental and computational data elucidating the effect of solvation on the stability and the bond strength of dative bonds. The authors find that both stability and bond strength for dative bond increase with the dielectric constant of the solvent. This is an important finding that does not seem to be known. It differs strongly from the behavior of normal covalent bonds. It is laudable that the authors make a clear distinction between the bond energy or stability and the bond strength which is independent of the fragments. The paper should be published after a revision addressing these issues:

Reviewer’s Point #1:

It is rightfully pointed out that "bond strength" cannot be determined directly, and is also not very well defined. They also point out that stability and bond strength do not necessarily correlate. Unfortunately, the manuscript is not really clear on which measures are used to assess bond strength. One good measure is the force constant of the bond, or the frequency, which is found in Table 2. The NMR J-coupling is the first measure given in Tab. 1. The authors write (p.4) "It thus reflects chemical bonding". Do they mean: "reflects bond strength"? Is there further evidence for this relation? If not what conclusion can be drawn from the NMR data?

In the conclusion, the "intrinsic energy" is stated as measure for "bond strength". This is not elaborated on in the Introduction or Results section.

Response: We have discussed this point in the revised manuscript. The following sentences are given in the revised manuscript, Introduction, pg 4

"It is thus related to chemical bonding (in contrast to direct coupling, which is a through-space interaction)³⁹ and is used to discuss the changes in B–N distances. However, the magnitude of the J-coupling can't be directly correlated to the bond strength."

Concerning the comment on the intrinsic energy as a measure of the bond strength, the relevant discussion is given in the Results and Discussion, pg 6:

"This provides arguments to discuss the correlation of the stability of dative bond complexes and the relevant bond strengths in terms ($\Delta(\Delta G)$, $\Delta(\Delta E_{inter})$) or ΔE^{solv} with changes in the intrinsic energies ($\Delta(\Delta E_{intrin})$) and force constant of the low-frequency vibrations, also listed in Table 3. Arguments for the validity of the use of ($\Delta(\Delta E_{intrin})$) values are based on the assumption that the low-frequency vibrations are mainly due to the B–N stretching with an almost constant contribution (65–67 %, Table 2) to the normal mode motion. Fig. 2 illustrates the dependences of all these characteristics and vibration frequency on the solvent polarity, showing almost exact matches in their trends."

Reviewer's Point #2:

The calculated data from Table 3 are very important for this work. Unfortunately, one can only guess from the Methods section what calculations are behind Table 3.

Response:

Detailed calculations of behind Table 3 have been included in the methods section.

Reviewer's Point #3:

The English could be improved

Response:

The language was corrected by the native English-speaking chemist.

Summary:

The nature and properties of the $\text{Me}_3\text{N-BH}_3$ complex stabilized by $\text{N} \rightarrow \text{B}$ dative bond have been explored through FT-IR, H-NMR, and B-NMR measurements and computational studies based on DFT and coupled cluster calculations and molecular dynamic simulations.